# The Important Role of Global State for Multi-Agent Reinforcement Learning

Shuailong Li [1,2,3] , Wei Zhang [1,3,]*, Yuquan Leng [4,5,]* and Xiaohui Wang [1,2,3]

1    State Key Laboratory of Robotics, Shenyang Institute of Automation, Chinese Academy of Sciences, Shenyang 110016, China; lishuailong@sia.cn (S.L.); wangxiaohui1@sia.cn (X.W.)
2    Institutes for Robotics and Intelligent Manufacturing, Chinese Academy of Sciences, Shenyang 110169, China
3    University of Chinese Academy of Sciences, Beijing 100049, China
4    Shenzhen Key Laboratory of Biomimetic Robotics and Intelligent Systems, Department of Mechanical and Energy Engineering, Southern University of Science and Technology, Shenzhen 518055, China
5    Guangdong Provincial Key Laboratory of Human-Augmentation and Rehabilitation Robotics in Universities, Southern University of Science and Technology, Shenzhen 518055, China
*    Correspondence: zhangwei@sia.cn (W.Z.); lengyq@sustech.edu.cn (Y.L.)

**Abstract:** Environmental information plays an important role in deep reinforcement learning (DRL). However, many algorithms do not pay much attention to environmental information. In multi-agent reinforcement learning decision-making, because agents need to make decisions combined with the information of other agents in the environment, this makes the environmental information more important. To prove the importance of environmental information, we added environmental information to the algorithm. We evaluated many algorithms on a challenging set of StarCraft II micromanagement tasks. Compared with the original algorithm, the standard deviation (except for the VDN algorithm) was smaller than that of the original algorithm, which shows that our algorithm has better stability. The average score of our algorithm was higher than that of the original algorithm (except for VDN and COMA), which shows that our work significantly outperforms existing multi-agent RL methods.

**Keywords:** multi-agent reinforcement learning; environmental information; deep reinforcement learning

## 1. Introduction

With the development of deep learning [1–5], reinforcement learning, and deep learning are combined and better-developed [6–10]. AlphaGo [11] is a method based on deep reinforcement learning and Monte Carlo tree search. It has defeated the top professional chess players among human beings and attracted extensive attention world-wide. AphaGo Zero [12] defeated AlphaGo. AlphaGo Zero proves the powerful ability of deep reinforcement learning and will help to promote the further development of artificial intelligence represented by deep reinforcement learning. With the development of research, deep reinforcement learning has been applied to many perceptual decision-making problems in complex high-dimensional state space. Among these, multi-agent cooperation is an outstanding representative of this complex problem.

In multi-agent reinforcement learning (MARL) [13–19], one of the challenges is how to represent and use the action-value Q function of multi-agent [20,21]. In the decision-making of multiple agents, we learn the strategy of each agent for decision-making and learn a centralized behavior value function to evaluate the joint action. MARL needs to consider other agents and their environmental information. This shows that environmental information is more important in MARL.

Independent Q-learning (IQL) [22] abandons a centralized behavioral value function and learns an independent behavioral value function independently. However, this method

cannot clearly represent the interaction between agents and may not converge. Counterfactual multi-agent (COMA) [23] learns a fully centralized state–action value function Q_tot and uses it to guide the optimization of decentralized policies in an actor–critic (AC) framework. However, this method is likely to lead to low sample efficiency, and when there are many agents, this method becomes impractical. Value decomposition networks (VDNs) [24] represent Q_tot as a sum of individual value functions Q_a that condition only on individual observations and actions. However, VDN severely limits the complexity of centralized action–value functions. QMIX [20] ensures that a global maximum performed on Q_tot yields the same result as a set of individual maximum operations performed on each Q_a.

These methods make decisions based on a partially-observable environment while ignoring global environmental information. An observation o is a partial description of a state, and some information may be omitted. However, a state s is a complete description of the state of the world. Reinforcement learning controls an individual who can act independently in a certain environment and continuously improve its behavior through interaction with the environment. In reinforcement learning, agents always make decisions in a certain global state. Therefore, environmental information directly determines the decision-making of agents. Environmental information is particularly important. Different from these methods, our work fully considers environmental information rather than only local information when making decisions.

Because observation o is a part of environmental information, it is not best for agents to make decisions only through action–observation history [25]. Using only observation to make decisions will miss environmental information. In multi-agent decision-making, agents make decisions according to their observations, and need to cooperate with other agents. This situation requires the participation of environmental information.

To avoid the disadvantage of using observations only in previous research methods, we pay attention to the importance of the global state and integrate the global state into decision-making. We evaluate our algorithm on a range of unit micromanagement tasks built in StarCraft II. Our experiments show that our improved algorithm is better than the original algorithm.

## 2. Materials and Methods

A. Multi-Agent Reinforcement Learning Methods

A MARL can be defined by a tuple $G = \langle S, U, P, r, Z, O, n, \gamma \rangle$. $s$ ($s \in S$) refers to the state in a true environment. At each time step, each agent selects an action $a$ ($a \in A \equiv U^n$) to form a joint action $u$ ($u \in U \equiv U^n$). When a joint action $u$ is selected, the agent will reach a new state after interacting with the environment. This process is represented by the state transition function $P(s'|s, u) \colon S \times U \times S \to [0, 1]$. $r(s, u)$ is the reward function and $\gamma \in [0, 1]$ expresses a discount factor.

A partially observable scenario represents the environment observed by a single agent, not all the information in the environment. Each agent has an episode $\tau$, and the episode is represented by action–observation pairs or action–state pairs. The joint action is obtained from the joint policy $\pi^a(u^a|\tau^a)$ and has a joint action–value function, $Q^\pi(s_t, u_t) = E_{s_{t+1:\infty}, s_{t+1:\infty}}[R_t|s_t, u_t]$, where $R_t = \sum_{i=0}^{\infty} \gamma^i r_{t+i}$ is the discounted return.

Deep Q-learning [25–30] is a classic deep reinforcement learning method. Deep Q-networks (DQNs) use a deep neural network to represent the action–value function $Q(s, a)$. In DQNs, the training data are the stored transition tuple $\langle s, u, r, s' \rangle$ and the squared TD error is used to train the parameters $\theta$:

$$L(\theta) = \sum_{i=1}^{b}\left[\left(y_i^{DQN} - Q(s, u; \theta)\right)^2\right] \tag{1}$$

where $y^{DQN} = r + \gamma max_{u'}Q(s', u'; \theta^-)$. $\theta^-$ is the parameter of a target network periodically copied from $\theta$ and remains unchanged for a number of trainings.

In MARL, agents are trained based on action–observation history. However, this method cannot fully consider the overall factors of the environment. The work of this paper takes the global state into account in strategy training, hoping to improve the effectiveness of strategy.

In this section, we describe our approaches for adding environmental information to multi-agent settings.

In the RL community, real-time strategy (RTS) games have always been an area of concern. Some games, such as football games and StarCraft [23], offer a great opportunity to tackle competitive and cooperative multi-agent problems. Competitive and cooperative problems are the main concerns in behavioral decision-making. Solving these problems is also the main aim of RL. Here, our algorithm was applied in StarCraft. Thanks to its rich set of complex micro-actions, which allow the learning of complex interactions between collaborating agents, StarCraft has been widely used in the RL community. Previous work applied RL to the original version of StarCraft: BW [31]. We performed our experiments on the StarCraft II Learning Environment (SC2LE) [32], which is based on the second version of the game and has been supported by the developers of the game.

B.    Environmental importance in MARL.

In this work, we focus on the important role of environmental factors in RL. An important feature of reinforcement learning is to learn in constant interaction with the environment. Multi-agent reinforcement learning needs to cooperate according to different environmental states. In this case, the learning of the environmental state is particularly important. Attention to this detail prompted our engagement with this work.

Independent Q-learning (IQL) decomposes a multi-agent problem into a collection of simultaneous single-agent problems that share the same environment. Value decomposition networks (VDNs) aim to learn a joint action–value function, and each agent conditions only on individual action–observation histories. QMIX estimates joint action–values as a complex non-linear combination of per-agent values. QTRAN proposes a new value decomposition algorithm. COMA uses the global critical network to evaluate the Q value and uses the non-global actor network to determine the behavior. However, these algorithms do not fully consider global information when making decisions, and we wanted to add environmental information representing global information to decision-making.

The simplest way to apply the environmental information to multiple agents is to input the state information into the algorithm. Because the state information is high-dimensional, dimensionality reduction is required. Our method integrated the state into the algorithm, making full use of the state information. In experiments, our method was compared with the QMIX, QTRAN, IQL, VDN and COMA algorithms. To better illustrate our method, this paper takes the QMIX algorithm as an example. The application of our method in other algorithms is similar to that in the QMIX algorithm. The network structure of QMIX is shown in Figure 1.

As shown in Figure 1, the framework of QMIX is composed of two parts; the top part obtains the action and the bottom part obtains the Q value. In the process of obtaining actions, the input of the framework is the number of agents, the action space, the observation space of agents, and the state space of the environment. The dimension of the observation space is 80, the dimension of the action space is eleven, the number of agents is five and the dimension of the state space is 120. The process of the framework is described below. The top part, which obtains the action, is composed of a splicing layer, two fully connected layers, and a recurrent neural network. The algorithm uses the splicing method to preliminarily process the input and obtains $5 \times 216$ dimension data. The output of the first fully connected layer is 64 hidden units. The name of the recurrent neural network is GRU. The output of the first fully connected layer is eleven hidden units. The actions are obtained by random sampling through the categorical function.

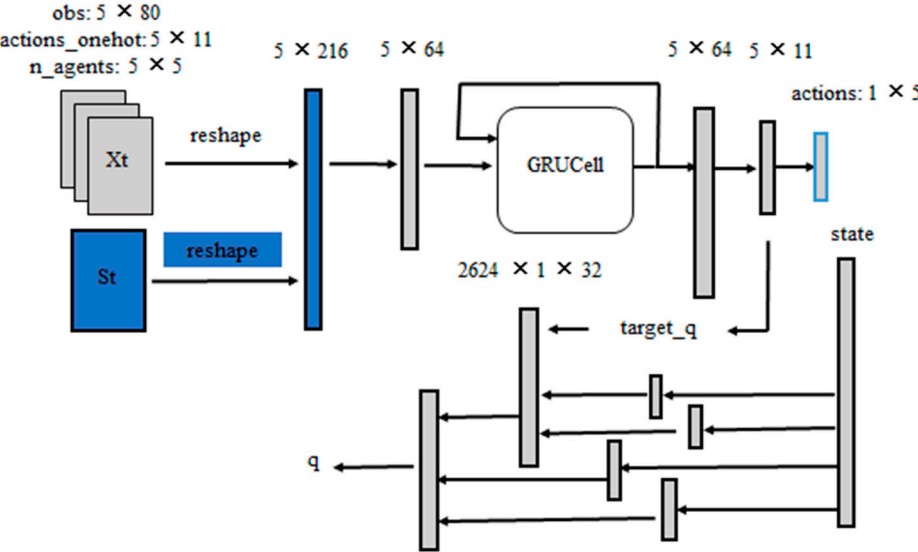

**Figure 1.** The framework of QMIX algorithm. Our algorithm takes environmental information (the blue part in the figure) as input to affect environmental decision-making.

For the top part, the dimension of input is $5 \times 216$, which is the sum of the dimensions of action space, observation space, number of agents and state space. To obtain actions, we used two fully connected layers and a recurrent neural network (GRU). The data processing was as follows. The first layer is a fully connected layer with an output of $5 \times 64$. The second layer is GRU, whose output is $5 \times 64$. The third layer is a fully connected layer with an output of $5 \times 11$. Then, we can obtain actions through the sampling process. To obtain the Q value, the bottom part uses an encoder model with two fully connected layers. To obtain the weights of the two fully connected networks, both consist of two fully connected layers and one activated layer, and the bias of the first layer is a fully connected layer. To ensure that the weights are non-negative, an absolute activation function follows the weights of the two fully connected networks. The process is as follows. The inputs of the first layer are the state and the target_q, the output of which is 32. The state has been processed through the weight network. The inputs of the second layer are the state and the output of the first layer, the output of which is 1. The state has been processed through the weight network. Figure 1 illustrates the feed-forward neural network.

To introduce environmental information into agent decision-making, the state is added to the initial input of the algorithm. In this case, the algorithm can fully integrate the information obtained by the agent with the environmental information and then make better decisions. The feed-forward neural network is the influence of the joint behavior of agents in the environment state, and the algorithm uses the action–value Q function to express the influence. To express this influence, the algorithm combines the joint behavior with the environmental state, so that the decision-making of the agents can be better evaluated.

QMIX is trained end-to-end to minimize the following loss:

$$L(\theta) = \sum_{i=1}^{m} \left[ (y_i - Q(\tau, a, s; \theta))^2 \right] \tag{2}$$

where $m$ is the batch size of transitions sampled from the replay buffer, $\tau$ is a joint action-observation history, $a$ is a joint action and $s$ expresses the environment state. $y_i = r + \gamma max_{a'} Q(\tau', a', s'; \theta^-)$ and $\theta^-$ are the parameters of a target network, as in DQN.

To better explain our work, the parameters of the algorithm are shown in Table 1. As seen from Table 1, our parameters are the same. The purpose is to better compare the algorithms. It can be seen from the algorithm parameters that we added only environmental information to the algorithm.

**Table 1.** The parameters of the algorithm.

| Algorithm | QMIX | QTRAN | IQL | VDN | COMA |
|---|---|---|---|---|---|
| Action_selector | Epsilon_greedy | Epsilon_greedy | Epsilon_greedy | Epsilon_greedy | Epsilon_greedy |
| agent | rnn | rnn | rnn | rnn | rnn |
| agent_output_type | q | q | q | q | q |
| Batch_size | 32 | 32 | 32 | 32 | 32 |
| Batch_size_run | 1 | 1 | 1 | 1 | 1 |
| Buffer_cpu_only | true | true | true | true | true |
| Buffer_size | 5000 | 5000 | 5000 | 5000 | 5000 |
| Critic_lr | 0.0005 | 0.0005 | 0.0005 | 0.0005 | 0.0005 |
| env | Sc2 | Sc2 | Sc2 | Sc2 | Sc2 |
| Epsilon_finish | 0.05 | 0.05 | 0.05 | 0.05 | 0.05 |
| Epsilon_start | 1 | 1 | 1 | 1 | 1 |
| gamma | 0.99 | 0.99 | 0.99 | 0.99 | 0.99 |
| Grad_norm_clip | 10 | 10 | 10 | 10 | 10 |

## 3. Results

To verify our experiment, we compare this algorithm with other classical multi-agent algorithms; the results of the training process are shown in Figure 2. As can be seen from Figure 2, the effect of the COMA, QTRAN, and VDN is the same as that of the original algorithm. The IQL algorithm is obviously better than the original algorithm. The QMIX algorithm is slightly better than the original algorithm. It can be seen from the training process that our work can obtain good results based on the original algorithm by only considering the factors of the global state. The results show that the global state plays an important role in MARL.

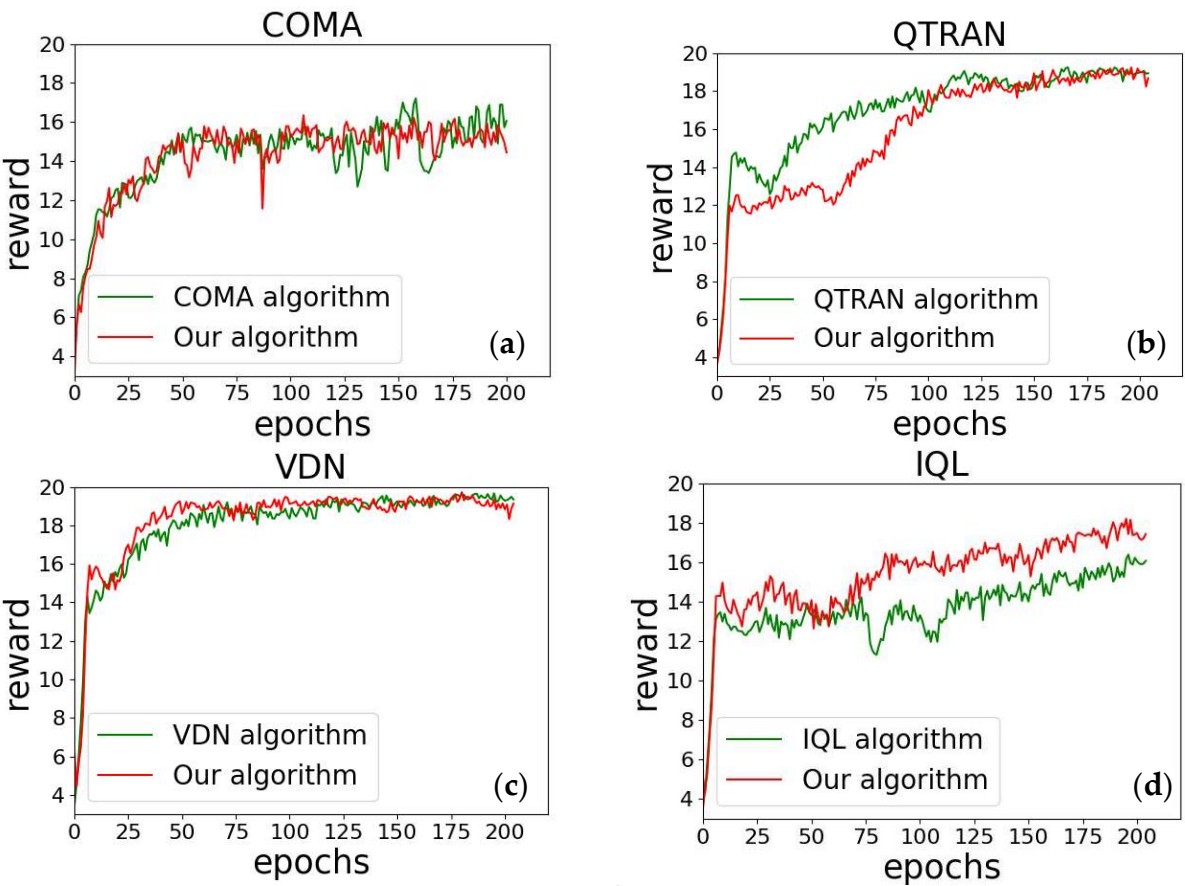

**Figure 2.** *Cont*.

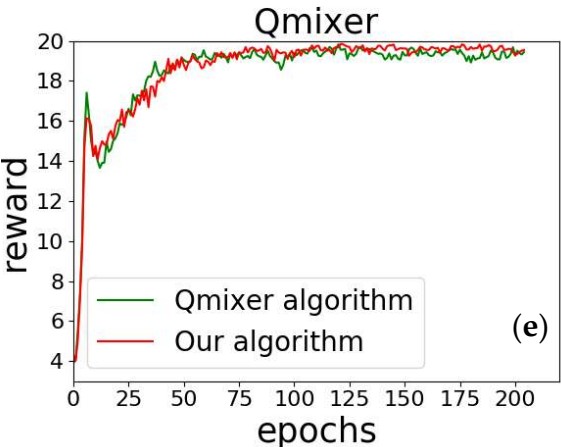

**Figure 2.** Comparison chart of training process. Our algorithms are compared in COMA algorithm (**a**), QTRAN algorithm (**b**), VDN algorithm (**c**), IQL algorithm (**d**) and Qmixer algorithm (**e**).

In Figure 2, our algorithm is compared with the original five algorithms. Our algorithm only adds the global state to the input of the original algorithm. In the training process, our algorithm can obtain the same or better effect as the original algorithm. Especially in the IQL algorithm, our algorithm obtains obviously excellent results. As seen from Figure 2, among other algorithms, our algorithm obtains better results in convergence and stability. To better verify our algorithm, we performed experiments in the test set.

Figures 3 and 4 show the test comparisons between our algorithm and the original algorithm after network training. Figure 3 shows the standard deviation of the test return, and Figure 4 shows the mean value of the test return. In QMIX, the standard deviation of our algorithm is 0.8856, and the standard deviation of the original algorithm is 1.22065. In IQL, the standard deviation of our algorithm is 2.6189, and the standard deviation of the original algorithm is 3.4538. In QTRAN, the standard deviation of our algorithm is 1.4837, and the standard deviation of the original algorithm is 1.4971.

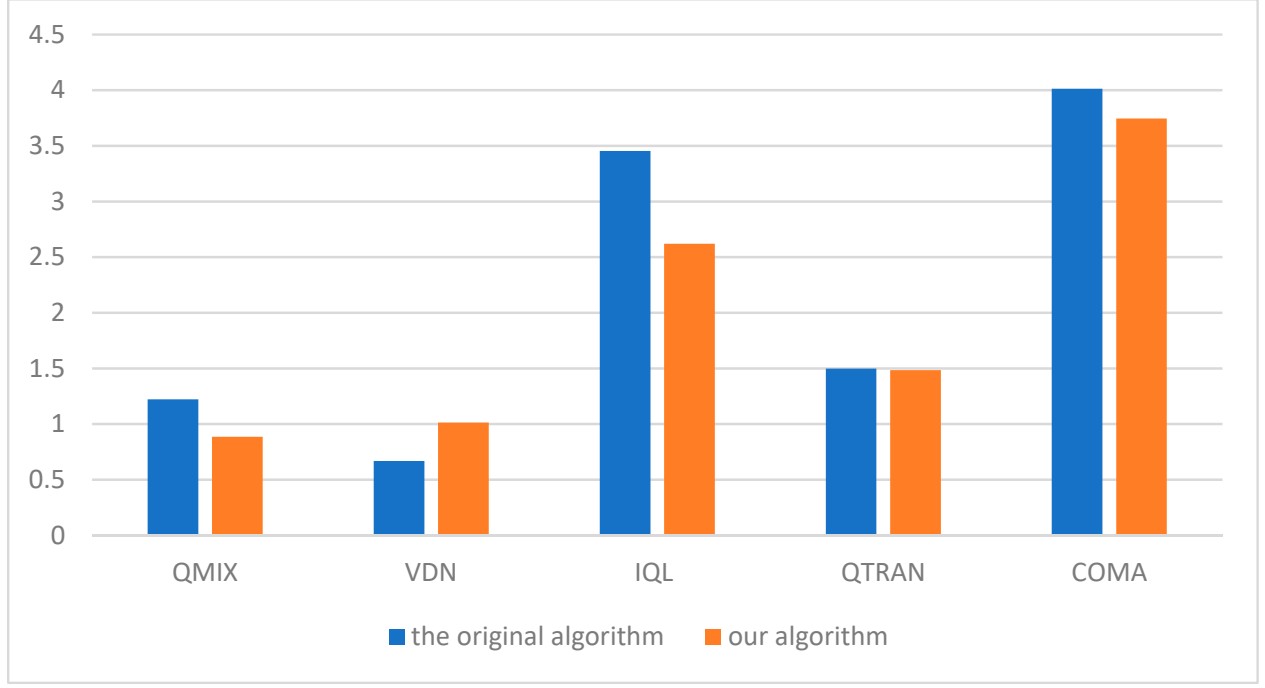

**Figure 3.** Standard deviation of test return.

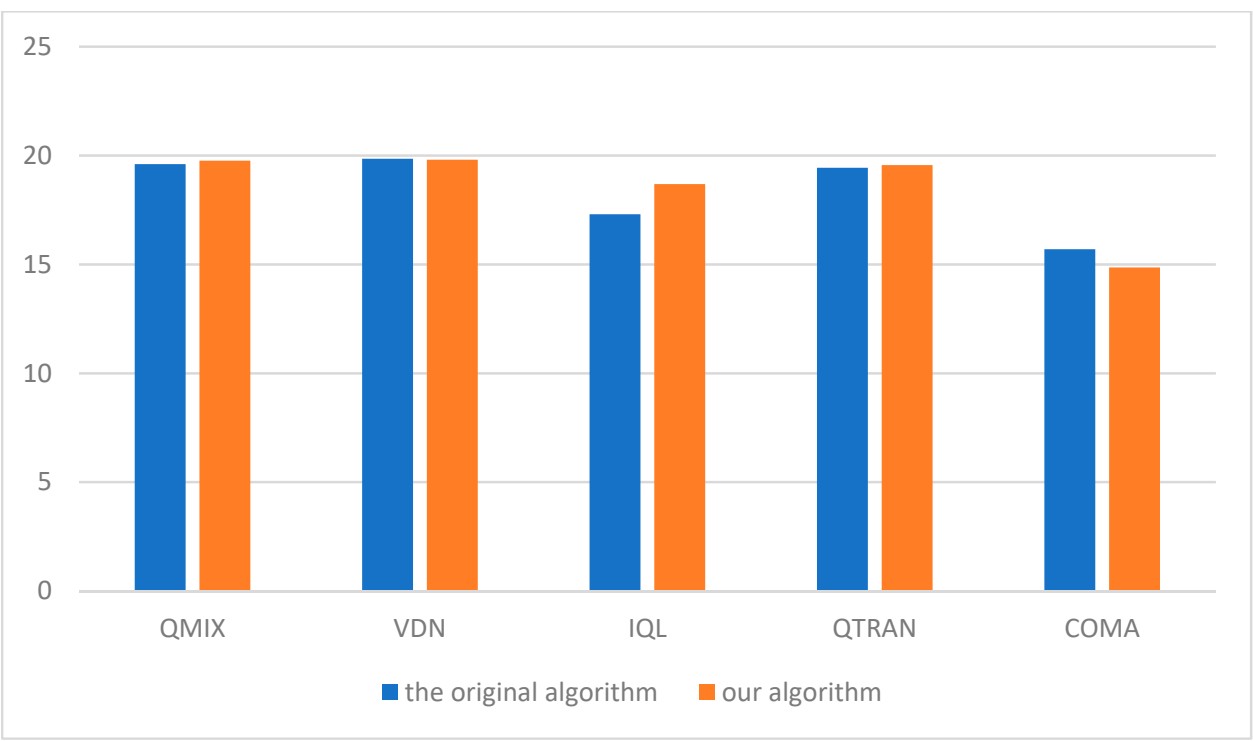

**Figure 4.** Mean of test return.

In COMA, the standard deviation of our algorithm is 3.7447, and the standard deviation of the original algorithm is 4.0122. However, in VDN, the standard deviation of our algorithm is 1.0129, and the standard deviation of the original algorithm is 0.6669. As seen from Figure 3, the standard deviation of our algorithm is smaller than that of the original algorithm (except the VDN algorithm), which shows that our algorithm has better stability. In QMIX, the mean return of our algorithm is 19.7573, and the mean return of the original algorithm is 19.6021. In IQL, the mean return of our algorithm is 18.6853, and the mean return of the original algorithm is 17.302. In QTRAN, the mean return of our algorithm is 19.553, and the mean return of the original algorithm is 19.4334. However, in VDN, the mean return of our algorithm is 19.8003, and the mean return of the original algorithm is 19.8477. In COMA, the mean return of our algorithm is 14.8588, and the mean return of the original algorithm is 15.6895. Figure 4 shows that our algorithm has a higher return than the original algorithm (except VDN and COMA), which shows that the effect of our algorithm is better than that of the original algorithm.

## 4. Conclusions

This paper emphasizes the important role of the global state in multi-agent strategy decision-making. Our algorithm added a global state to strategic decision making, and we compared it with several classic multi-agent algorithms. The experimental results show that our algorithm is effective and that our work is meaningful. This article can obtain better results solely by adding environmental information. Our work shows the importance of environmental information in reinforcement learning. In future work, MARL researchers need to fully consider environmental information.

However, our algorithm does not obtain good results in some algorithms (such as the VDN algorithm). To make the work more universal, we will apply the global state to the model-based reinforcement learning algorithm in our future work.

**Author Contributions:** Formal analysis, Y.L. and X.W.; Funding acquisition, Y.L. and W.Z.; Methodology, S.L. All authors have read and agreed to the published version of the manuscript.

**Funding:** This research received no external funding.

**Data Availability Statement:** Not Applicable, the study does not report any data.

**Acknowledgments:** This work was supported by the National Natural Science Foundation of China under Grant 52175272, 51805237 and Joint Fund of Science & Technology Department of Liaoning Province and State Key Laboratory of Robotics, China (Grant No.2020-KF-22-03).

**Conflicts of Interest:** The authors declare no conflict of interest.

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
