# Peer review of "The Important Role of Global State for Multi-Agent Reinforcement Learning"

_futureinternet, doi:10.3390/fi14010017_

Round 1

Reviewer 1 Report

First, I am not sure that the paper fits the mission and audience of this Journal.

Second, authors should make an effort to illustrate their model more effectively.

In particular, in order to make the presentation of their model clear, I suggest that authors revise both section 2 and section 3. Section 2 should be split into two sub-sections, the first one to illustrate their model and the second one to explain how it is implemented in the study to get results.

Please, emphasize how paper contributes to literature. An in-depth literature survey is necessary to clearly emphasize weaknesses and gaps that this paper wants to address.  

Figure 1 is unclear. Why do authors present the QMIX? Why using different colors in figure?

The choice of measurements of parameters used in the model should be motivated.

Please, discuss implications to industry and research of study results and the proposed algorithm. 

Author Response

Comment 1: First, I am not sure that the paper fits the mission and audience of this Journal.

Response: Thank you for pointing this out. Included topics of Future Internet are artificial and augmented intelligence, smart learning systems. We use reinforcement learning to solve the decision-making problem of multi-agent, which is a part of artificial intelligence and also belongs to the learning system. So our work fits the mission and audience of Future Internet.

Comment 2: Second, authors should make an effort to illustrate their model more effectively.

Response: Thank you for pointing this out. According to your suggestion, I have modified the paper to better explain the model. In our paper, the model is summarized in Sec. 2 Materials and Methods (Page 4); as follows:

“For the top part, the dimension of input is 5 x 216, which is the sum of the dimensions of action space, observation space, number of agents and state space. To obtain actions, we use two fully connected layers and a recurrent neural network (GRU). The data processing as follows. The first layer is a fully connected layer, whose output is 5 x 64. The second layer is GRU, whose output is 5 x 64. The third layer is a fully connected layer, whose output is 5 x 11. Then we can get actions through the sampling process. To obtain the Q value, the bottom part uses an encoder model with 2 fully connected layers. To obtain the weights of the two fully connected networks, both consist of two fully connected layers and one activated layer, and the bias of the first layer is a fully connected layer. To ensure that the weights are non-negative, an absolute activation function follows the weights of the two fully connected networks. The process is as follows. The inputs of the first layer are the state and the target_q, whose output is 32. The state has been processed through the weight network. The inputs of the second layer are the state and the output of the first layer, whose output is 1. The state has been processed through the weight network. Figure 1 illustrates the feed-forward neural network.”

Comment 3: In particular, in order to make the presentation of their model clear, I suggest that authors revise both section 2 and section 3. Section 2 should be split into two sub-sections, the first one to illustrate their model and the second one to explain how it is implemented in the study to get results.

Response: Thank you very much for your advice. According to your suggestion, I have divided section 2 into two sub-sections and made relevant modifications.

Comment 4: Please, emphasize how the paper contributes to the literature. An in-depth literature survey is necessary to clearly emphasize the weaknesses and gaps that this paper wants to address.

Response: Thank you for pointing this out. According to your suggestion, we increased the research on relevant literature and emphasized the importance of our work. The amendments are summarized in Sec. 2 Materials and Methods (Page 3); as follows:

“Independent Q-learning (IQL) decomposes a multi-agent problem into a collection of simultaneous single-agent problems that share the same environment. Value decomposition networks (VDNs) aim to learn a joint action-value function, and each agent conditions only on individual action-observation histories. QMIX estimates joint action-values as a complex non-linear combination of per-agent values. QTRAN proposes a new value decomposition algorithm. COMA uses the global critical network to evaluate the Q value, and uses the non-global actor network to determine the behavior. However, these algorithms do not fully consider global information when making decisions. We want to add environmental information representing global information to decision-making.”

Comment 5: Figure 1 is unclear. Why do authors present the QMIX? Why use different colors in the figure?

Response: Thank you for pointing this out. Qmix algorithm is one of our experiments. In order to illustrate the experimental process, we cite qmix algorithm as an example. Relevant explanations are summarized in Sec. 2 Materials and Methods (Page 3); as follows:

“In our experiments, our method is compared with the QMIX, QTRAN, IQL, VDN and COMA algorithms. To better illustrate our method, this paper takes QMIX algorithm as an example. The application of our method in other algorithms is similar to the QMIX algorithm.”

Comment 6: Please, discuss implications to industry and research of study results and the proposed algorithm.

Response: Thank you for pointing this out. The implications to industry and research of study results has been discussed. Relevant discussion are summarized in Sec. 4 Conclusions (Page 8); as follows:

“This article can obtain better results only by adding environmental information. Our work shows the importance of environmental information in reinforcement learning. In future work, MARL researchers need to fully consider environmental information.”

Reviewer 2 Report

The text requires significant English improvements. Many sentences are not appropriately formulated. The mathematics is not sufficiently detailed, and the formulas in the text must be numbered. It seems that more information is necessary to convince on the application of the method. Results must be better demonstrated by including sufficient information on how the testing was set up. More information about the use case where the algorithm was applied must be included. Comparison of various traditional algorithms requires a preliminary description of these algorithms. Figure 1 must be better explained. Conclusions are poor. It seems that the overall paper is too short and requires significant expansion in order to make your contribution explainable to a higher detail.

Author Response

Thank you for your careful review of our manuscript. The comments are valuable for improving our paper. We have modified the article in light of the reviewers’ comments. We have addressed all of the issues mentioned in the reviewers’ comments and would like to submit the revised manuscript entitled The Important Role of Global State for Multi-agent for consideration to be published on Future Internet.

In our revised paper, all the modified parts are marked in RED color. To make sure all comments and our revisions are fully understood, we give a relatively detailed response to the reviewers’ comments (the original comments are marked with BOLD BLACK in this document). Detailed corrections and responses to the reviewers are as follows, quote of the manuscript which has been underlined and marked in BLUE color.

All of the changes in the manuscript have been marked in red, including:

  1. (Page 2-3) We reword the description “Different from these methods, our work fully considers environmental information rather than only local information when making decisions.”

“Independent Q-learning (IQL) decomposes a multi-agent problem into a collection of simultaneous single-agent problems that share the same environment. Value decomposition networks (VDNs) aim to learn a joint action-value function, and each agent conditions only on individual action-observation histories. QMIX estimates joint action-values as a complex non-linear combination of per-agent values. QTRAN proposes a new value decomposition algorithm. COMA uses the global critical network to evaluate the Q value, and uses the non-global actor network to determine the behavior. However, these algorithms do not fully consider global information when making decisions. We want to add environmental information representing global information to decision-making.”

to express the advantages of our work.

  1. (Page 3) We reword the sentences “In our experiments, our method is compared with the QMIX, QTRAN, IQL, VDN and COMA algorithms. To better illustrate our method, this paper takes the QMIX algorithm as an example. The application of our method in other algorithms is similar to the QMIX algorithm.” to explain why we use Figure 1.
  2. (Page 4) We reword the sentences “For the top part, the dimension of input is 5 x 216, which is the sum of the dimen-sions of action space, observation space, number of agents and state space. To obtain ac-tions, we use two fully connected layers and a recurrent neural network (GRU). The data processing as follows. The first layer is a fully connected layer, whose output is 5 x 64. The second layer is GRU, whose output is 5 x 64. The third layer is a fully connected layer, whose output is 5 x 11. Then we can get actions through the sampling process. To obtain the Q value, the bottom part uses an encoder model with 2 fully connected layers. To obtain the weights of the two fully connected networks, both consist of two fully connected layers and one activated layer, and the bias of the first layer is a fully connected layer. To ensure that the weights are non-negative, an absolute activation function follows the weights of the two fully connected networks. The process is as follows. The inputs of the first layer are the state and the target_q, whose output is 32. The state has been processed through the weight network. The inputs of the second layer are the state and the output of the first layer, whose output is 1. The state has been processed through the weight network. Figure 1 illustrates the feed-forward neural network.” to explain our model.
  3. (Page 2-4) We improved the structure of the article to make it more readable.
  4. (Page 8) We add the sentence “This article can obtain better results only by adding environmental information. Our work shows the importance of environmental information in reinforcement learning. In future work, MARL researchers need to fully consider environmental information.” to explain the importance of our method.
  5. We polished the article to make the article more readable.

Comment 1: The text requires significant English improvements. Many sentences are not appropriately formulated. The mathematics is not sufficiently detailed, and the formulas in the text must be numbered. It seems that more information is necessary to convince on the application of the method. Results must be better demonstrated by including sufficient information on how the testing was set up. More information about the use case where the algorithm was applied must be included. Comparison of various traditional algorithms requires a preliminary description of these algorithms. Figure 1 must be better explained. Conclusions are poor. It seems that the overall paper is too short and requires significant expansion in order to make your contribution explainable to a higher detail.

Response: Thank you for your recognition and helpful advice. We have polished the paper and numbered the formulas. To better describe the algorithm, we modify the relevant information of the paper. In future work, we will supplement our algorithm. Our revisions to the paper have been marked above in which has been underlined and marked in BLUE color.

Round 2

Reviewer 1 Report

The authors have satisfactorily addressed my concerns. Their paper can be published.